# Parent Perceptions of an Early Childhood System’s Community Efforts: A Qualitative Analysis

**DOI:** 10.3390/children10061001

**Published:** 2023-06-02

**Authors:** Brandie Bentley, Tuyet Mai Ha Hoang, Gloria Arroyo Sugg, Karen V. Jenkins, Crystal A. Reinhart, Leah Pouw, Ana Maria Accove, Karen M. Tabb

**Affiliations:** 1School of Social Work, University of Illinois at Urbana-Champaign, Champaign, IL 61801, USA; brandie2@illinois.edu (B.B.); lpouw3@illinois.edu (L.P.); 2All Our Kids Early Childhood Networks, Chicago, IL 60607, USA; anamaria.accove@illinois.gov

**Keywords:** early childhood development, community systems, parent perspective

## Abstract

Understanding how parents, and other primary caregivers, perceive and experience early childhood programs and services is essential for identifying family-centered facilitators and barriers to service utilization. Therefore, this paper aims to explore parent knowledge of and experiences with community efforts of an early childhood system in Illinois: the All Our Kids Early Childhood Networks (AOK Networks). Our research team conducted focus group interviews with 20 parents across four Illinois counties. A semi-structured interview guide was used to examine parent perceptions of an early childhood system’s community efforts in promoting the health and well-being of children aged from birth to five. Thematic network analysis was used to analyze all focus group data. Parents indicated three salient themes, including: (1) comprehensive information sharing practices, (2) diverse service engagement, and (3) barriers to service access. Overall, parents reported general satisfaction with the quality of available services and provided feedback regarding identified areas of need to increase the accessibility and utilization of local services. Engaging parents as partners is essential to the effective implementation of family-centered early childhood services. Families are the experts of their lived experiences, and incorporating their voices in program development and evaluation efforts works to increase positive child and family outcomes.

## 1. Introduction

Considerable efforts have been directed towards addressing, reducing, and preventing health inequities for children and families [1,2,3]. Health inequities, characterized as unfair and preventable systemic differences in health status and the distribution of resources influenced by social condition, have a significant impact on early life experiences [4]. Both beneficial and adverse early childhood exposures present within a child’s natural environment contribute to widened disparities across the life course [1,2,3]. Factors including familial poverty, stress, poor nutrition, mental health conditions, community violence, and exposure to environmental toxins all alter childhood developmental trajectories [3,5]. Early childhood services and systems play a critical role in promoting optimal health and well-being [6]. To effectively address heath inequities and ensure equitable access to these services, it is essential to understand the multi-level barriers faced by families seeking to engage with them.

Children aged from birth to five require various services, including health care, education, and more, to support their long-term positive health, growth, and development [7]. Despite high importance, early childhood services are often disjointed, each operating in individualized siloes, leaving families overwhelmed and underserved [7]. Effective early childhood systems strive to remedy this issue, as they operate with a centralized goal of caring for children and their families by facilitating necessary connections to supportive programming and resources such as Early Intervention, Head Start, public health, and home visiting services [8,9].

Early childhood interventions have been proven to set the foundation for a cumulative advantage that extends across the life course by increasing families’ access to support and protective factors. Research also shows that they are most effective and cost efficient during the early childhood period [2,3,10]. Positive developmental outcomes established during early childhood serve as a foundation for long-term successes in health, language and communication, cognitive, and social-emotional development [11,12]. Additionally, risk factors often overlap and require an integrated approach that recognizes families’ and children’s complex needs for support. Therefore, equitable access to high-quality early childhood services benefits individuals, families, and communities at large.

Well-functioning early childhood systems are comprehensive and integrate diverse stakeholder groups to advocate for child and community health. Parents and other primary caregivers, direct service providers, and local and state policymakers each offer unique insights into the necessity, quality, and accessibility of local services. However, when assessing the overall effectiveness of community-level early childhood system development efforts, there is a limited focus on parent perspectives and experiences. Attention is more often drawn towards the experiences of professional service providers and specialty populations represented within the system, such as teenaged parents and parents of children with disabilities [9,13]. 

The All Our Kids Early Childhood Networks (AOK Networks) is a state-wide collaboration funded by the Illinois Department of Human Services. The AOK Networks connect leadership from direct service sectors and community partnerships under a shared goal of supporting children from birth through age five, their caregivers, and expecting parents. Collectively, the AOK Networks’ members utilize shared resources and expertise to address child and family issues at the local and systemic levels. There are 11 AOK Network counties in Illinois that engage representatives from various early childhood services, including, but not limited to, health, mental health, early learning, social services, faith-based organizations, and parent support. Each AOK Network engages families as valued partners in their mission of establishing and maintaining an accessible, equitable, and just local early childhood system. Ultimately, the AOK Networks work to maintain a coherent early childhood system that aims to provide protection against multiple risk factors affecting health, development, and social-emotional outcomes.

Understanding how diverse parents and primary caregivers perceive and experience the offerings of their early childhood system is essential to accurately identify facilitators and barriers to service utilization, especially amongst underserved populations [13]. Further, highlighting this perspective also underscores the vital role parents have in shaping early childhood outcomes, and their importance in the conversation on early childhood health. Integrating parent feedback into program development and evaluation efforts increases accessibility to family-centered services and ensures that families are connected to the services they need to thrive. Therefore, additional research is needed to explore and integrate parents’ perceptions of local early childhood systems to guarantee the provision of accessible and aligned programming that reflects community interests. Thus, our qualitative study sought to understand parent perceptions of the AOK Networks community efforts by exploring their knowledge of available services, local programming experiences, and engagement benefits. 

## 2. Materials and Methods

### 2.1. Recruitment and Sample 

Convenience sampling methods were used to recruit and enroll focus group participants across four of the eleven counties: Adams, Kane, Tazewell, and Wabash/Edwards. We selected the four counties based on their distinct regional classifications (Adams—Micropolitan, Kane—Chicago Metro, Tazewell—Downstate Metro, and Wabash/Edward—Rural). The diversity of these regions captures a range of community contexts and allowed for a broad and representative study of parental experiences across the AOK Networks. AOK Network Coordinators from the four selected counties assisted with recruitment efforts by connecting our research team to leadership interacting with parents involved in local early childhood programming, such as play groups, parent ambassador meetings, Head Start parent groups, and Women, Infants, and Children (WIC) programs. Eligible study participants met the following inclusion criteria: (1) English or Spanish fluency, (2) parent or primary caregiver of at least one child aged 0–5, (3) currently participating in early childhood programs within their respective community.

The study sample consisted of 20 participants across the state of Illinois. Participant demographic characteristics are included in Table 1. The sample population represented a diverse makeup, with individuals self-reporting the following racial/ethnic categories: seven Non-Hispanic White (35%), four Non-Hispanic Black (20%), three Hispanic (15%), and six unknown (30%). Educational attainment varied, with the majority of the participants (60%) having a high school diploma or lower. Lastly, the age distribution was relatively wide-ranging. Participant ages ranged between 17 and 52, with more than half of the participants (60%) under the age of 30.

### 2.2. Data Collection

Between October 2018 and February 2019, our evaluation team conducted four in-person focus group interviews with 20 participants in the following AOK Network counties across Illinois: Adams (*n* = 3), Kane (*n* = 4), Tazewell (*n* = 9), and Wabash/Edwards (*n* = 4). All focus group interviews were conducted in English, and an on-site Spanish Interpreter was present for interviewees who spoke English as a second language for the interviews in Kane County. All participants provided informed consent and permission for the focus group to be audio recorded. Participants received a USD 5 gift card for their engagement. 

Trained focus group facilitators (BB, CR, and KT) used a semi-structured interview guide to explore parental knowledge, experiences, and perceived benefits of early childhood resources within their respective communities (Appendix A). Sample questions included “In your opinion, how good are the early childhood programs (or services) in your respective community? Has this changed over time?”, “Why do you participate in early childhood events or activities?”, and “Are there any areas of early childhood programs that need improvement?”. 

To accommodate the busy schedules of the parents and caregivers participating in our study, we strategically conducted our focus group discussions immediately after pre-scheduled parent meetings. By aligning our focus groups with existing commitments, we hoped to foster an environment that was both convenient and comfortable for our participants. The focus group interviews lasted 12–24 min each in duration, with an average interview time of 17 min. Due to the smaller focus group sizes, this was sufficient time for each participant to respond to our questions and provide additional information when follow-up questions were asked. An individual note-taker was present during each focus group session to document the discussion. 

### 2.3. Data Analysis

Our team used thematic network analysis to analyze all focus group data [14]. First, the audio files and field notes were transcribed verbatim. Following transcription, all team members independently reviewed each transcript in full to identify basic codes. The team then met and organized the basic codes into potential categorical themes. Authors BB, CR, and KT independently reviewed each transcript, highlighted themes across the transcripts, and met weekly to combine notes and identify common themes across each focus group. Lastly, the team created a graphic depiction of the thematic network analysis, including themes and subthemes. Individual images were compared to reach a consensus regarding the final themes. Figure 1 provides a graphic depiction of the thematic analysis.

## 3. Results

Three organizing themes emerged across the focus groups, including: (1) comprehensive information sharing practices, (2) diverse service engagement, and (3) barriers to service access. Within the organizing themes, eight basic themes were present: (1) peers, (2) agencies and organizations, (3) media, (4) prenatal and postpartum care, (5) cross-sector services, (6) transportation, (7) competing priorities, and (8) geographic location. 

Comprehensive Information Sharing Practices

Through comprehensive and overlapping information sharing practices, participants reported maintaining a sufficient awareness of available programs and resources to promote their child and family’s health. Participants shared that they relied on multiple forms of communication from their children’s early learning centers, peers, parent-centered Facebook groups, TV, radio, and community calendars to stay informed of local events and resources. For example, one participant described how direct service providers and a community social media group increased their awareness of upcoming events:
*“Honestly, I’ve been so impressed with every event they do, their communication is awesome because they just don’t give you a piece of paper, they have their Facebook group that you can look on or the family support workers. There’s multiple ways of communicating so you’re never left in the dark of what’s going on…”*(Adams County)


Other participants discussed receiving information from their child’s school and the local health department:
*“The schools are really good about telling different community events, too. I’ve really been grateful for that when they send home a flyer of like different businesses doing [local events] …”*(Adams County)
*“I learned about this parent group from [a representative] through the Health Office.”*(Wabash/Edwards County)

Impressed with the overall quality of community resources, a participant expressed their excitement about sharing news of local services and events with other caregivers:
*“I tell everybody—just because I’ve been so excited since it’s come here, and I haven’t been able to find these programs for the kids—like I keep extra handouts for the Quincy Park District to give to parents like, hey did you know about this event. I always tell them about stuff at Early Childhood Center if they have a kiddo here. The library stuff, farmer’s market, all that.”*(Adams County)

As an early childhood system, the AOK Networks’ stakeholders communicate important information to ensure families are connected to appropriate and timely resources. In turn, families feel supported, knowledgeable, and valued as vital members of the early childhood community.
2.Diverse Service Engagement

Participants discussed utilizing several diverse services for their children and families. Common programing and resources recommended by the study participants were wide-ranging and spanned to meet families’ unique needs across the early childhood period. Services discussed during the focus group fell into four core areas. First, federally funded programs such as WIC, Head Start, and Early Intervention. Second, birthing and caregiver support programs that offer doula and home visiting services, such as Good Beginnings, Hope Pregnancy, Women’s Pregnancy Center, and the Women’s Care Center. Third, childcare and early learning centers. Lastly, community-based resources including play groups, famers markets, local businesses that host family-centered events, and other offerings through the local park districts. Collectively, these services are cross-sectoral, target health, education, and social service needs, are often low- or no-cost, and support the holistic well-being of children and their families. 

Many caregivers expressed long-term service engagement and high levels of satisfaction with existing services. One participant shared their recommendation for prenatal and parenting support organizations, noting the free services that made a difference for them during their prenatal and postpartum periods:
*“I highly recommend the Women’s Care Center because if you go you learn. Their Thursday night classes, when you’re pregnant you go, and you talk to someone. They’ll teach you about pregnancy, delivery, and then you get coupons you can use to get whatever you need for your baby... The Thursday night classes are less than an hour long and you can get free diapers and wipes and clothes. And the doulas. My friend…had a doula from the Good Beginnings program, and I had one from the Women’s Pregnancy Center downtown. It was the best choice ever. Must have when you’re pregnant.”*(Tazewell County).

Another participant also discussed the benefits of attending a prenatal and postpartum support class and the diverse activities and topics covered by the course:
*“[They host classes on] everything. Breastfeeding, safe sleeping, you have anything from getting pregnant all the way up to after you have your baby. They have stuff for grandparents, they have stuff for aunts, they have stuff for siblings and… they do discipline, a cooking class sometimes, craft classes.”*(Wabash-Edwards County).

The AOK Networks recognize families’ needs for cross sector long-term services and provide fluid service continuity to best support local families. Across AOK Network communities, study participants expressed the multiple benefits of services that target all members of the family unit and provide support through increased education, awareness, and tangible resources. The AOK Networks maintain an effective safety net of services and support aimed at improving the lives of local families.
3.Barriers to Service Access

While those engaged in services reported positive experiences, barriers to service access were also commonly discussed in each of the focus group interviews. Competing family priorities for time and resources, access issues due to geographical location, and limited safe and reliable transportation options were complex barriers that impeded early childhood program engagement for some of the families. When discussing barriers, participants sometimes provided possible solutions to help increase accessibility. 

For example, one participant discussed their challenge balancing work and attending local early childhood meetings:
*“Work would be one barrier because a lot of families want to make sure that they take care of their family financially. It’s a challenge if I want to miss an hour of work compared to me having to go to a meeting.”*(Kane County).

As a potential solution, the same participant offered the following:
*“I feel like a lot of the information that the parents who cannot attend [miss] should still be provided just with maybe a flyer or something so that way you know what all they missed. At least they get a chance to see it when the kids get home.”*(Kane County).

Focus group participants in Wabash-Edwards shared outstanding community needs that had not yet been addressed at the time of the interview. Among the top priorities, participants discussed a need for local dental services that accept Medicaid health insurance for low-income families:
*“…the only thing that I really think about is probably still maybe a little bit missing is dental services for people that are on a medical card. There’s nothing locally. We’ve got Wednesday’s Child that will take care of emergency stuff or if a child is in pain, but just for cleanings and that kind of stuff, there’s not anybody local that they can go to.”*(Wabash-Edwards County).

For some participants, the closest dental services that accepted their insurance were hours away. In these cases, safe and reliable transportation was an added difficulty families faced:
*“A lot of people may have transportation that’s okay for them getting to work or close by, but when you start talking travelling 5 h in a car with your kid, their vehicles may not be in a shape that [is] safe transportation to get them there and back…”*(Wabash-Edwards County).

By understanding the daily barriers faced by families seeking early childhood services, the AOK Networks can use this information to strategically tailor program approaches and service delivery. 

## 4. Discussion

To our knowledge, this is one of the first studies to explore participants’ knowledge of and experiences with local early childhood system efforts within the state of Illinois. Focus group participants across four AOK Network counties were asked to identify their personal experiences with programming and services, outstanding areas of need, and recommendations for improvement. Key findings indicate the value of information sharing, the accessibility of diverse services, and unique service barriers faced by parents. Our findings are of notable importance, as this study captures the experiences of parents from four distinct counties, each representing varied regional classifications. Moreover, results from our study align with international research, underscoring the importance of targeted interventions to address issues that impact child and family well-being on a global scale [15,16]. This highlights shared challenges faced by diverse populations and the universal need for effective early childhood community systems efforts. 

Within our study, participants spoke to the comprehensive information sharing practices observed within their communities. Access to multiple forms of information about local programming, services, and events contributed to participants feeling supported as empowered advocates for their child and family’s health. By fostering strong communication channels, the AOK Network stakeholders promote a sense of belonging and empower families with the knowledge and resources they need to navigate and thrive within the early childhood system. Participants also expressed an enthusiasm to share information about the high-quality early childhood system efforts with other family members and peers, ultimately increasing community engagement. These results align with past studies that found that positive social support from peers and direct service providers was a significant motivator in early childhood service engagement [9]. This finding can be used to enhance programming efforts seeking to increase parent involvement. As our study showed, parents and other caregivers are eager to engage with and share information about effective and accessible programing efforts. 

Additionally, prior research has shown that early childhood programs aimed at assisting both parents and children facilitate positive service experiences and increased program engagement [13]. Relatedly, in this study, participants utilized resources that existed across service sectors including education, health, parenting support, and more. Participants identified the family-centered approach of many services as a benefit, noting that this approach sufficiently addressed the concerns of the entire family unit, increasing their capacity to support their child’s growth and development. Notably, many of the services referenced by participants were low- to no-cost and extended a continuum of care across the prenatal and postpartum periods. This indicates an important focus on families’ continued needs for accessible and affordable care to ensure optimum outcomes. 

Lastly, participants discussed areas they felt were unaddressed by the early childhood system and provided recommendations for improvement. Identified barriers to service access included competing priorities for time and financial resources, a limited availability of necessary providers and services in rural geographic areas, and a lack of safe and reliable transportation. These results align with a previous study that identified practical issues such as conflicting work schedules, the frequency and timing of services, and program accessibility as obstacles to parenting program engagement [17]. The responses indicate a need for multi-level efforts to affect sustainable change in the early childhood system, especially for families residing in underprivileged areas. This finding can be used to support systemic efforts aimed at increasing the availability of accessible services and addressing service gaps within rural communities. By decreasing the number of barriers perceived by caregivers, early childhood systems reinforce the value of caregiver engagement. Fewer barriers also support families in receiving essential resources, ultimately reducing the number of at-risk children and families [18,19]. In sum, successful community system efforts much acknowledge and unpack the complex interplay between individual, organizational, environmental, and cultural factors that influence outcomes for local children and families [16]. 

### Limitations

While this study provides valuable insights, it is also essential to acknowledge its limitations. First, we utilized a convenience sample framework, limiting our sample to individuals who were actively involved in local early childhood programming or network engagement efforts at the time of recruitment. Our relatively small sample size may have limited the depth of discussion, and having larger sample sizes could provide further exploration of complex experiences and perspectives.

Second, our sample population does not represent the full list of AOK Network counties. Therefore, our findings are not generalizable to the entire AOK Networks collaboration, and variations in socioeconomic and other community contexts could impact the relevance of our findings to other regions. Our research also does not include an analysis of differences in perception and experience based on the number of children, relation to child, or county-level comparisons.

Third, data were collected in 2018 and 2019, before the onset of the COVID-19 pandemic. It is essential to note that the pandemic has drastically affected early childhood outcomes, and our findings may not reflect these recent changes. However, findings from this study provide a valuable baseline understanding of parents’ and caregivers’ perceptions about the need for a well-coordinated early childhood system before the global health crisis. Additionally, the barriers identified within our study—competing priorities, economic hardship, limited access to health care services, and lack of reliable transportation—have persisted, and in many cases were exacerbated by the pandemic [20]. 

Lastly, while there was a diverse representation of parents across ages, education levels, and racial and ethnic backgrounds, our sample was limited to majority female participants. Future studies may consider including non-maternal caregivers, including fathers, grandparents, extended family members, and guardians. 

## 5. Conclusions

This study adds to a limited existing body of literature that broadly explores caregiver perceptions of early childhood systems’ community efforts. Overall, caregivers reported general satisfaction with the quality of available services. Early childhood systems can benefit from the reported successes and feedback regarding identified areas of need to increase the accessibility and utilization of local services. As the study participants shared, access barriers are often complex and require multi-level efforts for sustainable change. Recommendations for addressing these barriers include advocating for policies and programs such as increasing the number of health care providers who accept public insurance, providing subsidies for transportation expenses, implementing mobile health care services, and leveraging technology. We also recommend conducting further research and evaluation to identify best practices for improving service accessibility. Families are the experts of their lived experiences, and incorporating their voices in research, program development, and evaluation efforts contributes to the effective implementation of family-centered early childhood services, ultimately increasing positive outcomes for all.

## Figures and Tables

**Figure 1 children-10-01001-f001:**
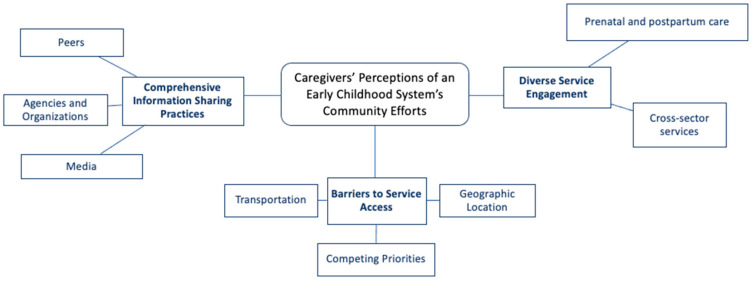
Thematic Network Analysis Results.

**Table 1 children-10-01001-t001:** Focus Group Participant Demographics.

Demographic	(n = 20)	n (%)
County	Adams	3
	Kane	4
	Tazewell	9
	Wabash/Edwards	4
Race/Ethnicity	Non-Hispanic/White	7 (35)
	Non-Hispanic/Black	4 (20)
	Hispanic	3 (15)
	Unknown	6 (30)
Education Level	Less than High School (8 years or less)	1 (5)
	Some High School (9–11 years)	5 (25)
	High School (12 years)	6 (30)
	Some College/Technical School (13–15 years)	3 (15)
	College Graduate (16 years)	1 (5)
	Graduate School (17 years or more)	3 (15)
	Unknown	1 (5)
Age	<19 years	3 (15)
	20–29 years	9 (45)
	30–39 years	3 (15)
	40–49 years	2 (10)
	50+ years	1 (5)
	Unknown	2 (10)

## Data Availability

In accordance with privacy regulations, ethical considerations, and the confidential nature of the data used in this study, the dataset is not publicly available.

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
