# Peer review of "Parent Perceptions of an Early Childhood System’s Community Efforts: A Qualitative Analysis"

_children, 2023, doi:10.3390/children10061001_

Round 1
Reviewer 1 Report
The title of the article is apt and correct. The abstract meets the criteria for this section, being concise and clear in the keyword count.
The introduction briefly provides a theoretical introduction to the topic, but provides it from only a limited number of literature sources (11). The methodology and sample selection is sound, clearly described and vividly presented. However, the data was collected in 2017-2018, which is older data and may no longer be relevant.
The discussion contains few literature sources, but this is understandable due to the originality of the topic. In the conclusions, I recommend the addition of options for removing the identified barriers or ways to address or explore them.
The limitations of the study are also a challenge for further studies and investigations in the field.
Author Response
Thank you for providing feedback on the different sections of our article. We appreciate your thorough review and would like to address each of your comments:
Additional up-to-date references have been included throughout.
A statement addressing the timeframe of our data collection was added to the limitations section (lines 389-396).
We added a few recommendations for addressing access barriers within the conclusions section (lines 408-416).
Reviewer 2 Report
Parent Perceptions of an Early Childhood System’s Community Efforts: A Qualitative Analysis
I enjoyed reading with well-written paper. The points I raise are minor ones that the authors will be able to easily address.
Introduction
A beautifully constructed and written Introduction to your paper, presenting a lot of evidence and concepts in a clear and accessible way.
Materials and Methods
It would help if you explain how/why you selected the four counties (out of the 11 counties). I note later in the paper you provide this explanation (Discussion, lines 277-279) but this needs to be clearer in the methods section.
Would the criterion of ‘English fluency’ disadvantaged/marginalised the voices of some of the population that AOK is aiming to reach? I appreciate your sample is diverse.
I note that your data were collected 2018-2019 and I absolutely understand that some papers take time to come to fruition. However, a lot has happened since then (pandemic, some increase in austerity… at least that’s the position in the UK, rising food and energy prices etc.). I’,m not suggesting that this time lag is a reason for not considering this paper for publication as it offers a lot but I think later in the paper in the Discussion and/or Limitations this ‘time since data collection’ perhaps needs to be contextualised. I assume from the way you write the introduction that AOK is still going strong?
The focus groups seem to be quite short. Again, not a problem per se, but maybe you can explain why they were quite contained in terms of time. This is probably important to address to allow you to reassure the reader that you went ‘deep’ in the focus groups, rather than just skittered across topics superficially.
Findings
Theme 1: I note that all the quotes, lovely as they are, are from Adams county. It would be ‘better’ to have quotes representing experiences from some other counties as well.
Quote 2 in Theme 2 (line 216): there’s a lovely typo (at least I assume it’s a typo). I think you mean ‘aunts’ not ‘ants’?
In Theme 3 you have 2 quotes from Kane county and two from abash-Edwards county. Again, I’d suggest making more of your data by swapping out one or two quotes and replacing them with quotes from Adams and Tazewell.
Discussion
I think your Discussion does a good job of contextualising your findings but maybe you can show a bit more clearly, perhaps in opening paragraph that your Illinois findings resonate with work done internationally. This could perhaps be gently woven through the Discussion. How similar are your findings to other networks/interventions/community-based efforts?
Please also consider the comment made earlier about contextualising your work that reflects the situation from 5 years-plus ago to a readership looking at this paper in 2023.
References
Quite a few of your references are a bit dated. I appreciate that seminal references can be a little dated and that some fields have a small evidence based but only one reference is from the 2020s. There is a lot of contemporary literature about social determinants of health, health disparities, inequality in childhood, parents, family engagement. I’m not suggesting that you delete all your references but I think you need to dive into the literature again and add some more up-to-date references.
Other
Funding statement (line 348) is incomplete.
Author Response
Thank you for providing feedback on the different sections of the article. We appreciate your thorough review and would like to address each of your comments:
Materials and Methods: We explained the selection of the four counties further within the Recruitment and Sample paragraph (lines 96-100).
We updated our inclusion criteria to read “English or Spanish fluency” as our sample also included Spanish-speaking individuals (lines 104-105).
A statement addressing the timeframe of our data collection was added to the limitations section (lines 389-396).
We provided additional detail regarding our focus group length in the data collection (lines 140-147) and limitations sections (lines 373-375).
Findings: We appreciate your suggestion to diversify the representation of participant quotes across all counties in Theme 1 & 3. We included an additional quote from Wabash/Edwards (lines 223-224) under Theme 1.
Thank you, we updated the word to “aunts”.
Discussion: Additional citations were added to the discussion.
References: Additional up-to-date references have been included throughout.
Other: The funding statement has been updated.